# Multi-Sensor Integration and Machine Learning for High-Resolution Classification of Herbivore Foraging Behavior

**DOI:** 10.3390/ani15070913

**Published:** 2025-03-22

**Authors:** Bashiri Iddy Muzzo, Kelvyn Bladen, Andres Perea, Shelemia Nyamuryekung’e, Juan J. Villalba

**Affiliations:** 1Department of Wildland Resources, Quinney College of Natural Resources, Utah State University, Logan, UT 4322-5230, USA; juan.villalba@usu.edu; 2Department of Mathematics and Statistics, Utah State University, 3900 Old Main Hill, Logan, UT 84322, USA; kelvyn.bladen@usu.edu; 3Department of Animal and Range Sciences, New Mexico State University, Las Cruces, NM 88003, USA; arperea@nmsu.edu; 4Division of Food Production and Society, Norwegian Institute of Bioeconomy Research (NIBIO), PB 115, N-1431 Ås, Norway; shelemia.nyamuryekunge@nibio.no

**Keywords:** random test split, cross validation, Random Forest, XGBoost, behaviors classification

## Abstract

This study used random test split (RTS) and cross-validation (CV) machine learning data partition methods to test different models to classify cattle behavior, including activity and posture states as well as foraging behaviors, using GPS coupled accelerometer data with 12 h/days continuous recording observation as supporting ground truth. RTS in XGBoost performed best for general activity state classification, while CV in Random Forest excelled in more detailed foraging behaviors and foraging behavior-by-posture classifications. Key movement indicators like speed, Actindex, and sensor values (x, y, and z) were vital in predicting behaviors, suggesting specific sensors for tracking behaviors of interest to ranchers. The results highlight the benefits of continuous monitoring and advanced data analysis for real-time livestock tracking, leading to better grazing management and more sustainable land use.

## 1. Introduction

Herbivores, particularly livestock (cattle), have an important role in the structure and functioning of arid and semiarid grassland ecosystems characterized by low and unpredictable rainfall. These ecosystems, often dominated by grass monocultures, are ecologically vital and yet vulnerable [1], often facing challenges such as limited forage availability, nutritional quality, and seasonal variations in plant productivity [2]. As primary consumers, herbivores are central to nutrient cycling, plant community dynamics, and overall ecosystem health. Their foraging behavior, including activities such as grazing, rumination, walking, resting, and drinking water, directly or indirectly influences vegetation structure, biodiversity, and soil health. The foraging activities of herbivores affect nutrient cycling and soil conditions, which are crucial for maintaining healthy plant communities [3]. For instance, optimal grazing can enhance nutrient availability by depositing dung and urine, which enriches the soil and supports plant growth [4]. However, overgrazing can lead to irreversible damage, such as soil erosion and loss of plant species diversity, ultimately decreasing land productivity [5]. It is also known that this subsequently changes the living conditions of wild animals [6,7]. Additional behaviors, such as grooming and nursing, are also important as they contribute to livestock populations’ social structure and overall well-being [8], indirectly affecting their productivity and ecological systems. Understanding these behaviors is vital for improving land management strategies and ensuring sustainability while mitigating the negative impacts of overgrazing and climate change [9]. Traditional methods of monitoring herbivore grazing behavior, such as fixed-point sampling and direct visual observation, are often impractical, particularly in the vast and remote locations throughout arid and semiarid grasslands.

Bailey et al. [10] recommended the use of precision livestock management (PLM) tools by real-time monitoring and management approaches that use global position system (GPS) tracking, accelerometers, and other sensor technologies in maintaining rangeland health. Nyamuryekung’e [11] emphasized the use of these animal movement tracking devices to assess grazing patterns and classify foraging behaviors in these challenging environments. While GPS collars provide precise location data, enabling researchers to track animal movements across large areas, additional data, such as identifying grazing hotspots across the landscape, are significant. For instance, Nyamuryekung’e et al. [12] used collars to provide insights into grazing, walking, and resting times, and other research groups (e.g., Brennan et al. [13]) have even assessed season-long livestock grazing behavior using low-cost collars. Accelerometers detect changes in movement patterns with specific behaviors, such as grazing and standing [14,15]. By combining GPS data with accelerometer information, researchers can more accurately infer foraging behaviors and better understand how herbivores utilize the landscape [16]. Bailey et al. [17] suggested that combinations of GPS tracking coupled with accelerometer monitoring may be more accurate than either device used by itself. However, Ganskopp and Johnson [18] pointed out that these devices have accuracy-related challenges like satellite-related errors and loss of satellite reception owing to atmospheric conditions, topography, canopy, and near infrastructure. Bonneau et al. [19] recommended combining telemetry devices with timestamp cameras, while Aquilani et al. [20] and Bailey et al. [17] further emphasized the use of both cameras with machine learning (ML) to enhance the assessment and understanding of livestock grazing behavior. Cameras can effectively capture foraging and mothering behaviors, such as grooming and nursing; social interactions, such as mating or fighting; and responses to environmental factors, such as vegetation changes, water availability, or the presence of predators. However, camera systems are subject to limitations, especially battery life, during continuous recording in free-ranging grazing systems with no power sources [21].

Several studies revealed that ML, a collection of powerful data processing and analysis techniques, can be applied to animal behavioral classification based on data collected using wearable sensors [14,22,23]. Integrating GPS, Accelerometer, and camera data requires advanced data analysis techniques, such as ML algorithms, to process large, multimodal datasets and automatically identify complex behavioral patterns. Supervised machine learning models have been used successfully to improve classification accuracy and effectively manage non-linear relationships in datasets, making them well-suited for distinguishing behaviors [24] like grazing versus non-grazing. For example, Augustine and Derner [25] studied grazing behaviors in semiarid rangelands using a two-axis accelerometer and GPS data, identifying distance traveled and head-down posture (assessed by the y-axis sensor) as primary indicators. They applied classification and regression tree (CART) analyses, achieving 87.8% accuracy for grazing and 86.5% for non-grazing activities, with an overall misclassification rate of 12.9%. Cabezas et al. (2022) [14] extended these findings by combining GPS and accelerometer data to analyze herd movement patterns and classify behaviors, thus demonstrating the added value of spatial data for sustainable rangeland use. Despite employing coupled devices, their study used different supervised ML methods for accelerometer-based behavior classification and unsupervised k-medoids ML algorithm for GPS data analysis.

In conclusion, a gap remains in integrating coupled GPS–accelerometer data with continuous behavioral observation systems, such as external field cameras, to test various supervised ML models for livestock behavior classification. This study hypothesizes that refining these models with more appropriate data partition techniques will lead to more reliable predictions, advancing precision livestock monitoring and promoting sustainable grazing management in diverse rangeland environments. While most studies have focused on confined systems, this research targets free-grazing environments, where landscape features and forage availability significantly influence cattle behavior and movement patterns. By continuously observing behaviors like grazing, rumination, and resting through external cameras, this study aims to improve the accuracy of behavior classification, which is crucial for optimizing grazing management and enhancing animal welfare. Accurate detection of these behaviors will also support better health monitoring by identifying signs of stress or discomfort. Furthermore, emphasizing the application of advanced ML data partition techniques, such as data splitting, cross-validation, and different supervised models, is expected to improve classification accuracy.

## 2. Materials and Methods

### 2.1. Study Areas

This study was conducted at the USU Richmond Research Farm (41.9227° N, 111.8136° W); elevation: 1511 m, annual temperature ranges −8.9 °C to 15.6 °C, annual precipitation of 525.8 mm and snowfall of 177.8 cm with Mendon silt loam soil type [26]. The experimental pasture, delineated by a five-strand barbed wire perimeter fence, entails 222,578.3 square meters of meadow brome grass (*Bromus inermis*) pasture monoculture (~2500 Kg/ha [27]) with an uneven distribution of intermediate wheatgrass (*Thinopyrum intermedium*). A semi-permanent electric fence was also built at the center of the pasture to divide it into 2 equal blocks. A temporary electric fence perpendicular to this fence divided the pasture into 6 paddocks of 36,421.74 square meters (Figure 1). The study procedures described herein were approved by the Utah State University Institutional Animal Care and Use Committee (approval number 2566).

### 2.2. Animals, Sensors, and Camera Deployment

The experiment was conducted from July to September 2024, as early summer offers peak forage quality, which declines through mid-summer into fall, providing an ideal period to observe cattle behavior under changing forage conditions that negatively impact animal productivity, for which numerous cow-calf improvement practices might be employed. Twenty-four Angus matured mother cows (body weight [BW] 614 ± 20 kg) and their nursing calves (6 to 8 months of age) (BW = 244 ± 4 kg) were randomly assigned to six paddocks of brome grass monoculture (4 pairs/paddock). Of these, 22 cows were randomly fitted with LiteTrack Iridium 750+ GPS collars coupled with triaxial accelerometers (Lotek Engineering, Newmarket, ON, Canada). Each collar weighed approximately 900 g and was designed for a neck size of 50 cm. The collars featured a buckle for easy placement, and the fit was adjusted to allow a person’s fingers to fit comfortably between the cow’s neck and the collar. The collars were set to collect one GPS position every 5 min. The Standard fix was collar configuration, recording 18 positions, which offers greater detail in tracking animal movements. In contrast, the SWIFT configuration, with only 9 positions, still provides adequate data to monitor foraging behavior at the same interval accurately. The reduced number of GPS positions in SWIFT fixes was offset by the method’s increased efficiency and lower power consumption. Data was transmitted via the Iridium satellite network. The collars operated within a frequency range of 148 MHz to 174 MHz for data transmission, ensuring reliable real-time communication with the satellite even in remote locations. They were designed to withstand extreme environmental conditions, with an operating temperature between −40 °C and 70 °C. The collars captured data such as GMT time (time of recording), latitude and longitude (geographic location), altitude (elevation above sea level), temperature (ambient temperature), voltage (collar battery level), DOP (Dilution of Precision, reflecting GPS data quality), and satellites (number of satellites used in the GPS fix).

Additionally, these GPS collars have incorporated a tri-axis accelerometer to track movement on three axes: X (left–right), Y (forward–backward), and Z (up–down), recording pendular beats across all axes every 5 min, coinciding with the GPS position fixes. The orientation of the collars was controlled during attachment to the animals to ensure consistent and reliable accelerometer readings across all axes, maintaining data consistency. Ground truth observations were conducted through video recordings using three GoPro Hero 12 cameras (GoPro Inc., San Mateo, CA, USA) from 27 to 29 August 2024, following an initial 12 h training session on 26 August. During this session, the cows were closely followed with cameras to familiarize them with the operator and equipment, ensuring their behavior was not affected during the recording period. Three cows were randomly selected as focal animals from the 22 collared cows, representing three of the six paddocks. To facilitate monitoring, these cows were marked on their backs with white paint (Rust-Oleum) for easy visibility from any angle. Three observers, each equipped with one of the GoPro cameras, were assigned to follow one focal cow per paddock. They maintained a 5 to 10 m distance to the animal to ensure comprehensive video recording during designated periods. Recording sessions were conducted over three consecutive days (27–29 August), with video footage captured using the cameras mounted with GoPro Volta for extended battery life. Each day, the cows were observed during three intervals: morning (7:00 am–11:00 am), afternoon (12:00 pm–4:00 pm), and evening (5:00 pm–8:00 pm). These intervals were chosen to capture the majority of the behaviors performed by the cows across daylight.

### 2.3. Pre-Processing of GPS, Accelerometer, and Camera Data

The experiment collected GPS and accelerometer data for all 22 cow collars (22GAD) (approximately 4 collars/paddock), which were pre-processed to ensure accuracy and consistency before analysis. Data pre-processing, cleaning, and visualization were conducted using ArcGIS Pro version 3.2.2 and R version 4.4.2. Using ArcGIS Pro, 22 GADS (latitude/longitude in degrees) were converted into a point feature class in WGS_1984 using the “XY Table to Point” tool, and points outside the study area were removed by clipping with the “Richmond study shapefile” for all paddocks. The GPS collars had a positional accuracy of approximately 10 m, which resulted in some data points crossing into adjacent paddocks, especially since the paddocks were close together and separated by an electric fence. This error caused cows’ data points to be detected in the wrong paddocks. To mitigate this, the “Buffer” tool in ArcGIS Pro was used to omit any points affected by this error by buffering a 10 m distance around the perimeter of each paddock. After that, the exported cleaned 22 GAD with GPS geographic coordinates (degrees) column was converted to a projected coordinate system of eastern and northern points (meters) in the Utah State geographical UTM 12 northern zone (WGS84) using R software. The timestamps, initially reported in GMT, were also converted to Utah Mountain Time (Zone-12, UTC-6), the local time zone for each GPS–accelerometer data point. We further adopted Liu et al.’s [28] data-cleaning procedure in our 22 GAD by omitting points with a duration of less than 5 min, at least 4 satellites used for data recording to estimate the position, and altitude ranges between 1500 and 1570 m based on the description of the study location. The dataset comprising merged GPS and accelerometer data for the observed three animals (3RC) was extracted from the overall 22 GAD dataset. During the cleaning process, only 1% of data points were omitted from the 3RC dataset. The distances traveled, speed, and tri-axis accelerometer (X: forward–backward, Y: side–side, and Z: up–down) showing movement patterns by a single cow was visualized in R. The data accurately reflected the cows’ movements, with high sensor spikes corresponding to active behaviors such as grazing and walking, while low spikes were associated with static behaviors.

Three trained technicians were assigned to watch each cow’s videos and label behaviors in the prepared datasheet. Five exclusive cow behaviors were distinguished: grazing, walking, resting, rumination, and drinking water. During the transcription process, a behavior change was considered when a cow shifted from one behavior to another, with the transition occurring at more than a 3 min interval. Grazing was defined as the cow actively foraging with the head down to the grass, with or without leg step movement; walking referred to cow displacement in space without foraging or ruminating; rumination was stationary behavior identified when the cow was observed chewing a regurgitated bolus of feed and continuing to chew until it was swallowed again; resting was defined as any period of immobility, either standing or lying down, with minimal movement, apart from head movements or chewing the cud, and when cows were not grazing, walking, or ruminating; and drinking water was defined as when the cow’s mouth was over or in the water trough. The cows’ resting or ruminating postures were also noted accordingly, as the animal was standing up or lying down. Nursing was defined as the behavior when a calf suckles from its mother while grooming referred to actions where cows lick, scratch, or use their heads or tongues to interact with themselves or other cows. An animal was labeled as performing a specific behavior if it engaged in the behavior for over 3 min. During transitions between behaviors, intervals lasting less than 3 min were considered part of the previous behavior, while intervals exceeding 3 min were classified as a change into a new behavior. These behaviors were further categorized into two primary states: active (AC) and static (ST). Active behaviors included walking, grazing, and drinking water, while static behaviors encompassed ruminating and resting. The cow’s standing or lying down posture was also recorded for each static behavior. Grooming and nursing were categorized as static behaviors when performed for prolonged periods but classified as “other” when occurring briefly. This structured approach to behavior categorization enabled detailed tracking of each cow’s behavior patterns and time allocation.

### 2.4. Feature Calculations

The accelerometer and GPS data were used to derive different features for input in several machine learning models to classify previously defined behaviors. The features calculated from accelerometer data were Sum_XYZ, calculated by aggregating the values of the X, Y, and Z axes, reflecting the overall intensity of movement, and Avg_XYZ was determined as the average movement values across the three axes. In addition, absolute values of each axis were computed using the absolute value notation (|X|, |Y|, |Z|), allowing for a more precise representation of movement intensity; Sum_XYZ_absolute involved summing the absolute values (|X| + |Y| + |Z|) of the X, Y, and Z axes; and Avg_Sum_XYZ_absolute was the average of these summed absolute values. The Activity Index (Actindex), conceptually resembling the Magnitude of Acceleration, was calculated by taking the square root of the sum of squared values of accelerations across the X, Y, and Z axes. This index measures the cows’ behavior levels, where higher values indicate more active behaviors and lower values suggest rest periods. Energy expenditure was computed as the cumulative sum of the Actindex, representing the total energy expended per duration of time particular behavior observed. We also adopted additional features from Versluijs et al. [29], such as overall dynamic body acceleration (ODBA), Vector of Dynamic Body Acceleration (VEDBA), Pitch, and Roll.(1)$$Pitch=artan(\frac{x}{\sqrt{y^{2}+z^{2}}})$$(2)$$Roll=artan(\frac{y}{\sqrt{x^{2}+z^{2}}})$$
*ODBA* = |*dx*| + *|dy*| + |*dz*|(3)
(4)$$Actindex=\sqrt{x^{2}+y^{2}+z^{2}}$$(5)$$VEDBA=\sqrt{{dx}^{2}+{dy}^{2}+{dz}^{2}}$$

From the GPS data, we calculated several metrics to analyze animal movement. Distance (meters) represents the distance traveled between two consecutive GPS points. Speed is derived from the distance traveled over time during the observation period, indicating the movement speed of the cow. The Straightness Index was calculated as the ratio of the straight-line distance between the initial recording and subsequent coordinates of a particular movement for each cow and day. Additionally, the Movement Angle (bearing) was determined using an arc-tangent function with respect to a due east trajectory. This calculates the angle (in radians) between the first and last coordinates, and this angle was subsequently converted to degrees. The derived GPS and accelerometer metrics were multiplied to calculate combined sensor features. This approach enabled a more comprehensive analysis of the cows’ movement and behavior patterns, providing deeper insights into their foraging behaviors using combined GPS–accelerometer devices. ML models were tested on different performance parameters using all calculated features to classify cattle foraging behaviors.

A Random Forest tree-based method was used to select features by aggregating trees and providing average feature importance. Following this, a filter method using the correlation coefficient was applied to measure the relationship between selected features and the target variable, helping to avoid feature redundancy. Thereafter, machine learning models were tested on different performance parameters using all calculated features to classify cattle foraging behaviors.

### 2.5. Data Partition Strategy and Machine Learning Models

The study applied the Random Train-Test Split (RTS) (70:30) ratioand 5-fold cross-validation (CV) data partition methods to evaluate various ML models for classifying animal activity and posture states, foraging behaviors, and foraging behavior-by-posture. RTS divides the dataset into training and testing subsets, typically using a 70:30 ratio, and is particularly effective in resource-constrained scenarios [30]. However, the split ratio can influence model performance, and multiple splits have been shown to enhance reliability [31]. In contrast, a five-fold CV partitions the data into five equal subsets, rotating the training and testing roles across all folds [32]. This method minimizes overfitting and provides a more robust performance estimate [33]. Despite the strengths of these approaches, challenges in model selection remain. As Cawley and Talbot [32] emphasized, biased performance evaluation can occur when selection criteria are improperly optimized, underscoring the need for careful model selection practices. The study evaluated six ML models for behavior classification, each employing distinct learning approaches: Perceptron adjusts weights based on errors, Logistic Regression predicts probabilities using a sigmoid function, Support Vector Machine (SVM) separates classes with optimal margins [31], K-Nearest Neighbors (KNN) classifies based on the nearest data points, Random Forest (RF) aggregates multiple decision trees for robust predictions, and XGBoost (XGB) enhances accuracy through gradient-boosted trees [32]. To ensure a fair and reliable assessment of both methods and their impact on classification accuracy, the study prioritized using the models with the highest accuracy in RTS as the benchmark for evaluating the CV method.

### 2.6. Models Performance Assessment

The performance of the models was evaluated based on several metrics. Accuracy was measured as the proportion of correctly classified instances (both positive and negative) out of the total instances: (TP + TN)/(TP + TN + FP + FN). Precision measures the proportion of correctly predicted positive instances out of all predicted positives: TP/(TP + FP). Recall estimates the proportion of correctly predicted positive instances out of the total positive instances: TP/(TP + FN). The F1 score combines the precision and recall scores into a single measure, providing a balance between the two: 2 × (Precision × Recall)/(Precision + Recall).

## 3. Results

### 3.1. Data Summary and Challenges

The data instances of different behavior classifications from the three observed mother cows are presented in Figure 1. Behavioral classification was based on GPS movement patterns, accelerometer thresholds, and direct observations. Some challenges occurred during data collection. Three GPS collars apart from the observed cows failed to collect data throughout the experimental period, which reduced the sample size. Camera overheating after 3 to 4 h due to high summer temperatures disrupted continuous video recording intended for validation.

### 3.2. Behavior Classification Using Random Train-Test Split Method

#### 3.2.1. Activity States Classification

The performance of different models is shown in Table 1. Various features were included as input to optimize classification accuracy across machine learning models. The features selected—X, Y, Z, Actindex, distance, and speed—achieved the highest classification accuracy. Ensemble learning models achieved higher classification rates and fewer misclassifications than simple classifiers. The Perceptron model exhibited the lowest performance among the models tested, with an accuracy of 63.8%, lower than Logistic Regression (72.4%). Similarly, SVM achieved an accuracy of 71.1%, which is 2.9% lower than KNN (74.0%). However, ensemble models outperformed other approaches, with Random Forest achieving 73.2% accuracy, 9.4% higher than Perceptron, along with consistent static metrics of 74.0%. XGBoost demonstrated the highest performance, achieving 74.2% accuracy, 10.4% higher than Perceptron, a static precision of 72.0%, a static recall of 79.0%, and a static F1 score of 75.0%. These results highlight the effectiveness of ensemble learning models for this classification, particularly XGBoost, in accurately classifying cows’ activity states. 

#### 3.2.2. Foraging Behaviors Classification

The same features used for state classification were applied, excluding Sum_XYZ after feature selection. Random Forest achieved the highest overall accuracy in general foraging behaviors, with accuracies of 65.9% for GR (grazing), 68% for RE (resting), and 50% for W (walking) (Table 2). It excelled in GR and RE but struggled with W. XGBoost followed closely, with 63.3% accuracy for GR, 67% for RE, and 13% for W, facing similar challenges with walking. SVM showed the highest precision for W (100%) but had low Recall (4%), indicating it could identify W when present but missed many instances. Logistic Regression performed poorly with W, with an F1 score of 0%, making it ineffective for classifying W. For fine foraging behaviors, Random Forest remained the most reliable model, with an overall accuracy of 59.7%, excelling in GR (67%) and RU (ruminating) (60%) (Table 2). XGBoost had an accuracy of 61.7% for GR and RU, but struggled with RE, similar to its performance with general foraging behaviors. In summary, Random Forest was the most consistent and accurate model for both general and fine foraging behaviors classification. XGBoost showed solid performance but had limitations with W and RE. SVM had high precision for W but poor recall, while Logistic Regression underperformed overall, especially with W.

### 3.3. Behavior Classification Using Cross-Validation Method

#### 3.3.1. Activity States Classification

We adopted a random 5-fold cross-validation technique to assess performance in activity classification. We used features selected from above to identify consistent predictors across methods for classifying activity states as static (ST) or active (AC). Performance metrics showed that XGBoost outperformed other models with 74.2% accuracy, followed by Random Forest at 73.2%, while Logistic Regression had the lowest accuracy at 69% and a 31% misclassification rate. XGBoost hyperparameters included 200 rounds, eta of 0.06, maximum depth of 4, colsample_bytree and min_child_weight of 0.5, and subsample of 0.8. Partial Dependence Plots (PDPs) revealed that speed and Actindex were the strongest predictors of AC, with Actindex increasing sharply up to 30 and stabilizing, and speed rising until 50 m/min before plateauing (Figure 2). X showed a slight positive effect, Y displayed a negative sigmoidal trend, and Z had a positive sigmoidal pattern. These results confirmed that speed, Actindex, and X were strongly associated with AC, with XGBoost delivering the most precise and confident predictions compared to the broader distributions in Logistic Regression and Random Forest.

#### 3.3.2. Foraging Behaviors Classification and by Posture

The features selected were consistent with those identified in the RS method, suggesting a stable pattern across both techniques of which the same set of features were used in foraging behaviors classification and by posture. Since XGBoost and Random Forest were found to perform better, we focused on these models for fine foraging behavior classification, including posture-based classification in the cross-validation method. After fine-tuning, XGBoost slightly outperformed Random Forest, achieving 69.38% overall accuracy compared to 68.51%. Both models performed well for grazing (GR, ~67%) and resting (RE, ~77%), but struggled with walking (W), where XGBoost (48.3%) outperformed Random Forest (22.2%) (Table 3). After reclassifying behaviors into GR, rumination (RU), and RE, accuracy dropped to 62.38% (RF) and 60.35% (XGB). For precision, Random Forest performed better for RE (47.1% vs. 36.5%), while XGBoost had a slight edge for GR (67.2% vs. 66.6%) and RU (58.9% vs. 59.6%). Both models performed well for posture classification (standing vs. lying down), with Random Forest at 83.94% and XGBoost at 83.7%. RF excelled in LD (79.9%), while XGB performed slightly better in SU (85.1%). For foraging behaviors-by-posture classification, both models had ~58.8% accuracy. XGBoost was better at detecting RE_LD (47.6%), while Random Forest performed better for RE_SU (46.2%).

The Partial Dependence Plots (PDPs) show that Actindex and speed are strong predictors for GR and W, with a sharp increase before stabilizing (Figure 3a). They decrease consistently for RE and RU, indicating a negative effect. X increases sharply for GR and plays a key role for RE and RU when it decreases (Figure 3b). Y shows more variability but is slightly linked to RU. Z drops sharply for RE and increases for GR. For posture classification (Figure 3c), speed and X are the strongest predictors of standing up (SU) over lying down (LD), with speed increasing sharply up to 50 before leveling off and X maintaining a positive trend. In posture-by-foraging behaviors classification, speed is a strong predictor of resting (RE_LD, RE_SU) and rumination behaviors (RU_LD, RU_SU), with lower values indicating these states. X and Z are secondary predictors, with higher X values linked to RE_LD (Figure 3d). In summary, speed is a key predictor for GR and W when they increase, and for RE and RU when they decrease. X is strong for GR when it increases, while both X and Y predict RE and RU when they decrease. Z predicts RE with a sharp drop and positively influences GR when it increases. Speed and X are the best predictors for SU over LD, emphasizing their role in classifying foraging behaviors.

## 4. Discussion

### 4.1. Activity State Classification

The classification of cows’ activity states provided key insights into the performance of the methods tested across various machine learning models. For classifying cows’ activity states (active vs. static), XGBoost achieved the highest accuracy, with 74.5% for RTS and 74.2% for CV. This consistency highlights the model’s robustness and aligns with prior findings by Li and Chai [34] and Ibrahim et al. [35], who noted XGBoost’s exceptional ability to handle complex datasets and accurately classify animal behaviors. Similarly, Random Forest also performed well, improving accuracy from 73.2% in RTS to 74.1% in CV. These findings are consistent with the observations of Chakraborty et al. [36], who demonstrated that ensemble models such as XGBoost and Random Forest effectively classify cattle behaviors using data from inertial measurement unit (IMU) sensor devices that track motion and orientation and closed-circuit television (CCTV) footage for manual annotation and validation. However, while Chakraborty et al. focused on monitoring cow health and estrus detection in a farm-based system integrating cloud infrastructure, our study specifically evaluates the models’ effectiveness in classifying cows’ activity states pertaining to grazing behaviors using a different dataset structure and experimental design. Our research emphasizes real-time behavioral classification in a production setting rather than health monitoring, highlighting the adaptability of these models in different contexts. Similarly, Wyner et al. [36] provided a theoretical perspective on ensemble learning methods, particularly the mechanisms behind AdaBoost and Random Forest. They propose that their success stems from their “spiked-smooth” classifier properties rather than conventional optimization principles. While our study aligns with their conclusions regarding the reliability of ensemble models, our findings extend these insights to the domain of livestock monitoring, demonstrating how such models can be applied effectively in real-world agricultural settings. However, despite the strong performance of ensemble models like XGBoost and Random Forest, challenges remain regarding scalability and robustness. Expanding trials to include more observations (cows) and integrating diverse data sources could further enhance these models’ performance [37,38]. Simpler models, such as Logistic Regression, were more prone to underfitting. This finding is evident in many PDPs in our present study, which reveal complex relationships between predictors and the response that a sigmoidal function cannot fully capture. Overall, ensemble models such as XGBoost and Random Forest consistently outperformed the simpler classifiers, showcasing their effectiveness in handling complex datasets with higher accuracy and fewer misclassifications.

Partial Dependence Plots (PDPs) revealed the relationships between key features and cows’ activity states, with speed and Actindex emerging as the most significant predictors of active behavior. These findings align with Li and Chai [34] and Mladenova et al. [39], who similarly identified these metrics as reliable indicators of cow activity. Additional X, Y, and Z features showed varying effects. Feature X exhibited a slight positive influence, Y displayed a weak negative trend, and Z followed a positive sigmoidal trajectory. Although not as strong as speed and Actindex, these features collectively contribute to the overall prediction accuracy [34]. Tracking speed and Actindex allow ranchers to identify active cows effectively while incorporating additional features like X, Y, and Z, which can further refine prediction accuracy. These insights offer actionable data for improving herd management by identifying key behavioral patterns associated with specific features [40]. Furthermore, the results underscore the value of non-parametric ensemble learning models like XGBoost, which demonstrated superior accuracy and robustness. By integrating ML models with meaningful feature selection, ranchers can develop more accurate and efficient livestock monitoring systems, ultimately enhancing animal behavior tracking and herd management practices [36].

### 4.2. Foraging Behaviors Classification

When comparing the performance of Random Forest (RF) and XGBoost (XGB) across different classification tasks, a clear distinction emerges between the cross-validation (CV) and random test split (RTS) methods. In the grazing (GR), resting (RE), and walking (W) classification tasks, the performance of XGBoost and Random Forest varied depending on the evaluation method. XGBoost slightly outperformed Random Forest using the cross-validation method, achieving an overall accuracy of 69.38%. However, in the random split method, Random Forest achieved a higher overall accuracy of 65.9%, compared to XGBoost’s 63.3%. Both models faced challenges with the walking (W) classification, but XGBoost achieved a higher accuracy of 48.3%, compared to RF’s 22.2%. This variability in performance aligns with findings from various studies that highlight the context-dependent nature of these algorithms. For instance, animal foraging behaviors can vary due to factors such as forage availability and climate. Walking (W) may be less prevalent in monoculture grassland systems than other behaviors, potentially leading to highly imbalanced data. The finding is supported by Wang et al. [41] who found that XGBoost outperformed Random Forest by better handling class imbalances and complex data structures, achieving higher accuracy. The present study differs from Wang’s group, which employed 10-fold cross-validation, while the present study used 5-fold cross-validation. The 10-fold approach is more robust, as it splits the data into 10 subsets, ensuring each data point is tested more frequently, reducing variability, and leading to more stable results. Additionally, Wang’s study relied on GPS data collected at a 1 min fix, whereas the present study used both GPS and accelerometer data with a 5 min fix. The longer sensor fix in the present study may also have contributed to the observed differences in model performance. Furthermore, XGBoost is particularly effective on structured, medium-sized datasets, as evidenced by its performance in telecommunications customer data analysis [42]. The results align with the grazing, resting, and walking tasks, where the dataset’s structure and size may have favored XGBoost in the cross-validation method.

Random Forest (RF) is known for its robustness and ability to handle medium datasets, which is particularly beneficial for classifying complex behaviors, such as those of cattle. When the foraging behaviors were reclassified into grazing (GR), ruminating (RU), and resting (RE), cross-validation with Random Forest achieved an overall higher accuracy compared to XGBoost (Table 3). Regarding precision, Random Forest demonstrated higher precision for GR and RU than for RE, indicating its effectiveness in distinguishing active behaviors like grazing and ruminating, while being less effective for resting. These results are consistent with the findings of Pütün and Yılmaz [43], who conducted their study on cattle grazing in free-range grasslands, a setting similar to the present study. Conversely, their study relied solely on accelerometer sensor data with a 1 s fix and utilized ground truth observations for training, collected from six cows over a noncontinuous recording period totaling 69 h. This resulted in a substantially larger dataset, of which our study represents only 1%. Additionally, they used only a random test split of 80:20, while our study applied a 70:30 split and 5-fold cross-validation. In contrast, our study utilized GPS and accelerometer data with a 5 min fix to classify grazing, ruminating, and resting behaviors, focusing on posture-specific classifications. However, Pütün and Yılmaz’s study also included additional behaviors such as mating, escaping, and licking mineral salt, which were not included in the present study. The algorithm’s overall accuracy in the present study was based on 5 min fixed-sensor data from three cows, continuously recorded over three days (48 h). Several studies on foraging behavior observation in long-range scenarios have reported higher accuracy, suggesting that integrating multi-sensor systems and ML for high-resolution data, as seen in continuous observations like the present study, could lead to even higher accuracy. Accuracy would likely increase further with extended observation periods over a week or two. Even in a confined dairy experiment conducted by Chen et al. [44], continuous observation provided a more precise representation of cattle behavior compared to fixed-interval sampling. Their study also highlighted that longer observation periods enhance accuracy, as behaviors fluctuate over time due to environmental factors, resource availability, and individual variability. After removing walking (W) and drinking water (DW), the focus shifted to the more distinguishable behaviors: grazing (GR), resting (RE), and ruminating (RU). RF continued to show superior performance in this updated classification, achieving higher accuracy compared to XGBoost (Table 2 and Table 3). The exclusion of walking and drinking water allowed for better differentiation between the remaining behaviors, reducing misclassifications. This improved performance highlights RF’s ability to better handle the task of distinguishing between complex behaviors like resting (RE), grazing (GR), and ruminating (RU), as seen in studies such as Wang et al. [25]. Several studies have achieved notable accuracy in classifying grazing and non-grazing behaviors. Augustine and Derner [25] reported 87.8% accuracy by observing five to nine cows during the summers of 2008 and 2011, with GPS position recorded at 5 min intervals and tri-axis accelerometers detecting 255 movements per interval. Similarly, Brennan et al. [13] achieved an 11.2% misclassification rate by observing 2–3 cows from 2016 to 2018 in South Dakota by using low-cost homemade GPS collars equipped with high-frequency three-axis accelerometers, recording GPS fixes at 1 min intervals and accelerometer data at 12 Hz. While these studies relied on intermittent observations over several years with extensive ground truth and high-frequency data, they are resource-intensive and less feasible for routine ranching. In contrast, the present study used continuous observations of three cows for 12 h per day over three days, with GPS and accelerometer data recorded at 5 min intervals. This approach achieved a classification accuracy of 74.5%, offering a less labor-intensive and more cost-effective and practical alternative with a longer battery life and affordable equipment for ranching operations. However, slightly less accurate, continuous observation at fixed intervals provides a solid foundation for real-time behavioral monitoring. Extending observation periods or increasing the sample size, combined with multi-sensor integration and ML, could further enhance the accuracy of foraging behavior classification, bridging the gap between resource-intensive research and practical applications for ranchers.

The Partial Dependence Plots (PDPs) revealed key features for predicting animal foraging behaviors. For grazing (GR), speed was the most significant predictor, with grazing probability decreasing as speed increased. This finding supports previous research by Jia et al. (2018), who noted that grazing cattle typically exhibit confined movement patterns. The Y-axis showed strong positive correlations, indicating lateral movements characteristic of grazing, while the Z-axis displayed a complex trend, reflecting the vertical head and neck motions during grazing. The findings are consistent with those of Sivakumar et al. [30], who observed similar head and neck movements during grazing. For resting (RE), speed and Actindex were the main predictors. Low speed and a stable Actindex indicated minimal movement, which aligns with findings from Liu et al. [24], who reported that resting periods in cattle are marked by low variability in movement data, showing minimal activity. In ruminating (RU), Actindex was the most influential predictor, capturing repetitive behavior patterns. The Y-axis showed moderate importance, with lateral movements correlating to rumination, and the Z-axis showed a pattern of vertical head motions, consistent with findings by Tamura et al. [45]. Additionally, speed played a moderate role, with limited but structured movement, while the X-axis contributed spatial movement data, supporting Zhang et al. [46], who found that the X-axis is likely to be relevant for sheep ruminating classification.

### 4.3. Posture and Foraging Behaviors-by-Posture Classification

In the posture classification task, where the model classified whether the animal was standing or lying down, RF again outperformed XGBoost, achieving high accuracy compared to XGBoost’s slightly lower accuracy. This result emphasizes RF’s strength in handling complex feature interactions, particularly in classification tasks that involve posture data. These findings align with Biau [47] who showed that RF excels in non-linear classification tasks, such as those involving subtle distinctions between postures and foraging behaviors. Kleanthous et al. [48] similarly found that RF performed well in tasks involving continuous sensor data with subtle distinctions, such as determining an animal’s posture. RF demonstrated superior performance across all sub-categories in the foraging behaviors by posture classification task. RF achieved high accuracy for grazing (GR), while XGBoost achieved slightly lower accuracy. However, both models struggled to classify resting (RE) when lying down (RE_LD) and standing (RE_SU), with RF achieving only moderate accuracy in these sub-categories. XGBoost performed slightly better in these sub-categories but still lagged behind RF in overall accuracy for this task. These findings are consistent with Chen and Guestrin [49], who noted that XGBoost performs well with balanced datasets but struggles with imbalanced class distributions, particularly in tasks involving fine-grained categories, such as posture-based foraging behaviors. In contrast, RF’s ability to handle multiple imbalanced categories makes it more robust. In conclusion, while both RF and XGBoost faced challenges, particularly with the classification of walking (W) and drinking water (DW), Random Forest (RF) consistently outperformed XGBoost across most tasks, especially in more complex classifications like posture classification and foraging behaviors by posture. RF’s ability to manage intricate feature interactions, particularly in tasks involving multiple categories or posture data, gives it a distinct advantage. XGBoost, although slightly more accurate in the Random Split Method after excluding problematic activities, still lagged behind RF in most tasks, particularly in posture and foraging behaviors-by-posture classification. These results align with previous studies, which suggest that RF is better suited for non-linear feature relationships, while XGBoost excels in linear tasks but struggles with high-dimensional and overlapping datasets. Additionally, the continuous nature of the observation data, combined with GPS and accelerometer sensors that record at 5 min intervals, could provide similar accuracy while extending battery life and reducing device costs. This makes it more feasible for long-term use in foraging behavior classification and addresses the challenges posed by highly variable and overlapping feature patterns. Understanding animal foraging behaviors becomes crucial for sustaining rangeland ecosystem health in arid and semi-arid regions, which are sensitive to disturbances such as overgrazing and desertification.

The Partial Dependence Plots (PDPs) revealed that speed and X-axis acceleration features are the strongest predictors for distinguishing between standing up (SU) and lying down (LD) in animal posture states. These predictors help to understand animals’ movement dynamics in facilitating posture transitions, as their variations provide critical insights into behavioral states [29]. Furthermore, in foraging behavior-by-posture classification, the X-axis and speed were also identified as important predictors for differentiating resting and rumination while lying and standing, reinforcing that GPS-coupled acceleration collars are pivotal in understanding animal behavior. The study highlighted that even using bi-axis accelerometer data, particularly the X and Z axes, could also predict posture states accurately. Liu et al. [28] highlighted that the use of multiple axes significantly enhances predictive accuracy in posture classification, as seen in the present study, where tri-axis accelerometer data improved posture detection models’ robustness and facilitated a more comprehensive understanding of animal posture and foraging behaviors by posture. Bailey et al. [17] further supported this perspective, indicating that integrating GPS with multiple axes accelerometer data is crucial for developing effective machine learning models that can accurately classify animal behaviors. These findings provide ranchers and researchers with viable options for selecting livestock monitoring devices, guided by the predictors highlighted in the present study, to accurately monitor animal behaviors while enhancing their management objectives, optimizing budgets, and meeting specific data requirements.

## 5. Conclusions

This study examined the use of random test split (RTS) and cross-validation (CV) data partition methods to test various models for classifying cattle behavior, including activity and posture states, foraging behaviors, and behavior-by-posture combinations, based on GPS and accelerometer data. XGBoost outperformed Random Forest (RF) in overall state classification (74.5% RTS, 74.2% CV). However, RF excelled in foraging behaviors tasks like grazing, resting, and walking (62.9% CV vs. 56.4% RTS) and more complex classifications such as posture (83.9% CV vs. 79.4% RTS) and foraging behavior-by-posture (58.8% CV vs. 56.4% RTS). RF’s ability to manage intricate feature interactions and imbalanced class distributions, particularly in walking and resting, made it more effective for these tasks. Partial Dependence Plots (PDPs) identified key features, such as speed and Actindex, which were essential for predicting active behaviors and grazing patterns, supporting previous research. The continuous sensor data collected at 5 min intervals over multiple days provided higher temporal resolution, capturing short-term behavior changes often missed in intermittent data collection methods. This approach improved classification accuracy and offered a cost-effective, practical solution for real-time livestock monitoring. Unlike multi-year, intermittent observation studies, continuous data collection allows for more detailed insights into behavioral shifts throughout the day. The ability to track behaviors like grazing, resting, and walking in real time makes this approach ideal for long-term monitoring and practical applications in livestock management. Additionally, accurately classifying and differentiating between activity states, foraging behaviors, and foraging behavior-by-posture are crucial for improving livestock management, particularly in free-ranging systems. Using machine learning algorithms like Random Forest and XGBoost in recommended data partition strategies based on the classification of interest livestock, managers can gain valuable insights into animal health, welfare, and productivity, enabling more informed decisions regarding resource allocation and pasture management. We recommend that ranchers and land managers adopt these methods, especially in systems using GPS and accelerometer data that can be set at 5 min fixed, as these sensors also improve monitoring precision, grazing management, health tracking, and resource distribution. Monitoring foraging behaviors through data-driven strategies can optimize pasture use and prevent overgrazing. Furthermore, future research should address data imbalances and sensor fix intervals to enhance classification accuracy. Integrating machine learning models with continuous behavioral data can revolutionize livestock management systems by offering real-time devices and accurate insights, enabling better decision-making, improving herd management, and boosting farm productivity. Future work should focus on scaling these methods, incorporating diverse data sources, and evaluating the long-term effectiveness of continuous monitoring for comprehensive livestock management.

## Figures and Tables

**Figure 1 animals-15-00913-f001:**
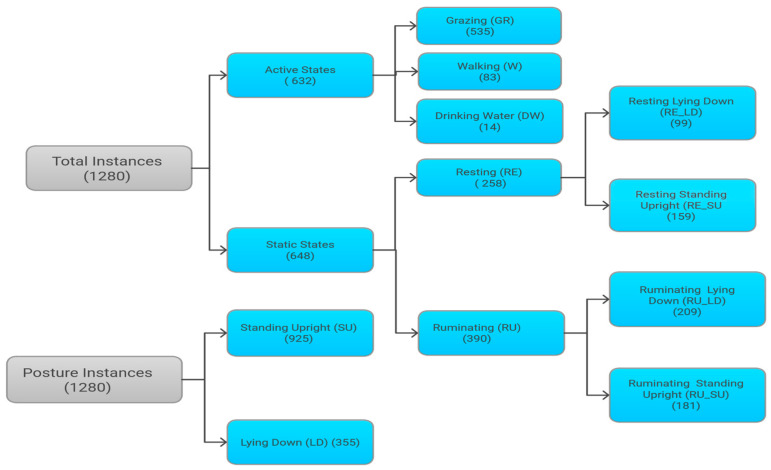
The dataset of 1280 recorded instances, with 632 classified as active and 648 as static. The foraging behaviors were distributed as follows: 535 cases of grazing (GR), 258 of resting (RE), 390 of ruminating (RU), 83 of walking (W), and 14 of drinking water (DW). Resting behavior was further broken down into 99 instances of resting in a lying down (RE_LD) posture and 159 instances of resting in a standing upright (RE_SU) posture. Ruminating behavior was similarly divided into 209 ruminating in a lying down (RU_LD) posture and 181 ruminating in a standing upright (RU_SU) posture. Posture data showed 925 instances of standing upright (SU) and 355 instances of lying down (LD).

**Figure 2 animals-15-00913-f002:**
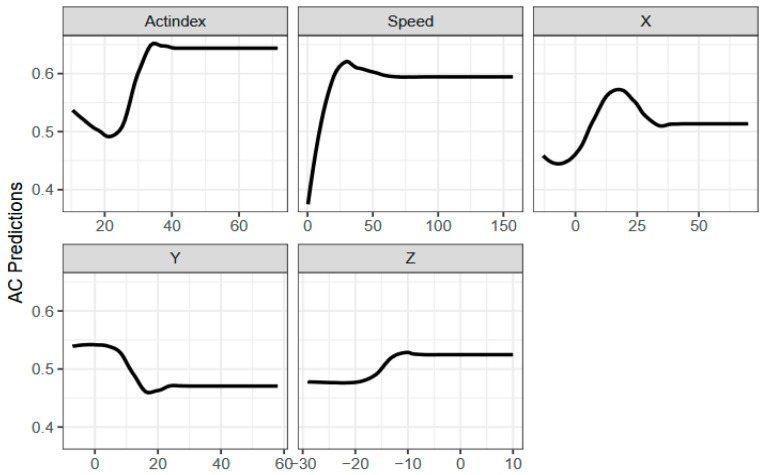
Partial Dependence Plots (PDPs) for predicting active states in cows using different features: X, Y, Z, speed (meter/minutes), and Actindex.

**Figure 3 animals-15-00913-f003:**
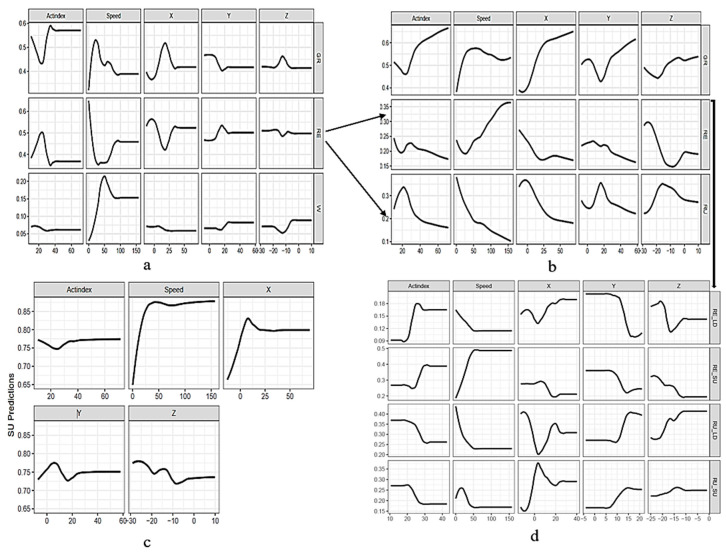
Partial Dependence Plots (PDPs) showing predicted features for (**a**) general foraging behaviors grazing (GR), resting (RE), and walking (W); (**b**) fine foraging behaviors GR, RE, and rumination (RU); (**c**) posture standing up (SU) versus lying down (LD); and (**d**) foraging behavior-by-posture, RE_LD (resting lying down), RE_SU (resting standing up), RU_LD (rumination lying down), and RU_SU (rumination standing). Features include speed (meters per minute), Actindex, and X, Y, and Z axes, analyzed using the CV method.

**Table 1 animals-15-00913-t001:** Table presenting performance metrics for each model on test data, including accuracy, precision, recall, and F1 score for active and static states using random test split method (RTS). The best results for a given metric are bolded to highlight which model was optimal for that analysis or task.

Model	Model Accuracy (%)	STATES					
		Active			Static		
		Precision (%)	Recall (%)	F1 Score (%)	Precision (%)	Recall (%)	F1 Score (%)
Perceptron	63.8	68	52	59	62	76	68
Logistic Regression	72.4	76	65	70	70	**79**	74
Support Vector	71.1	73	67	70	70	75	72
K-Nearest Neighbor	74	73	**74**	**74**	**74**	74	74
Random Forest	73.2	73	73	73	73	74	74
XGBoost	**74.2**	**77**	69	73	72	**79**	**75**

**Table 2 animals-15-00913-t002:** The table presents the overall performance of different ML models: grazing (GR), resting (RE), walking (W) and ruminating (RU) using random test split method (RTS). The best results for a given metric are bolded to highlight which model was optimal for that analysis or task.

Classification	Model	Foraging Behaviors	Precision (%)	Recall (%)	F1 Score (%)	Model Accuracy (%)
General foraging behaviors	Perceptron	GR	50	64	57	45.8
		RE	54	35	42	
		W	4	8	5	
	Logistic Regression	GR	60	53	56	61.2
		RE	62	76	68	
		W	0	0	0	
	SVM	GR	62	55	58	62.5
		RE	63	77	69	
		W	100	4	8	
	K-Nearest Neighbor	GR	55	64	59	60.4
		RE	65	65	65	
		W	100	4	8	
	Random Forest	GR	**63**	64	64	**65.9**
		RE	**68**	75	71	
		W	**50**	4	7	
	XGBoost	GR	63	62	62	63.3
		RE	67	72	69	
		W	13	8	10	
Fine foraging behaviors	Perceptron	GR	**70**	63	66	53.5
		RE	25	12	16	
		RU	46	68	55	
	Logistic Regression	GR	62	76	68	56.1
		RE	0	0	0	
		RU	49	66	56	
	SVM	GR	66	70	68	58
		RE	**53**	12	19	
		RU	50	**72**	59	
	K-Nearest Neighbor	GR	64	71	67	54.9
		RE	37	30	33	
		RU	50	50	50	
	Random Forest	GR	67	73	70	59.7
		RE	38	30	34	
		RU	**60**	62	**61**	
	XGBoost	GR	67	**78**	**72**	**61.7**
		RE	46	**31**	**37**	
		RU	59	59	59	

**Table 3 animals-15-00913-t003:** The table presents the overall performance of different machine learning models: grazing (GR), resting (RE), walking (W) and ruminating (RU) and resting in lying (RE_LD), resting in standing (RE_SU), ruminating in lying (RU_LD), and ruminating in standing (RU_SU) using the cross-validation method (CV). The best results for a given metric are bolded to highlight which model was optimal for that analysis or task.

Classification	Method	Model Accuracy (%)	Behaviors	Precision (%)	Recall (%)	F1 (%)
General foraging behaviors	Random Forest	68.51	GR	65.2	67.9	66.5
			RE	71.8	77.5	74.5
			W	22.2	2.4	4.3
	XGBoost	**69.38**	GR	67.1	67.2	67.2
			RE	72	77.9	74.9
			W	48.3	16.9	25
Fine foraging behaviors	Random Forest	**62.38**	GR	66.6	80.9	73.1
			RE	47.1	18.6	26.7
			RU	59.6	65.9	62.6
	XGBoost	60.35	GR	67.2	76.4	71.5
			RE	36.5	20.9	26.6
			RU	58.9	64.4	61.5
Posture	Random Forest	**83.94**	LD	79.9	47.7	59.8
			SU	84.7	96	90
	XGBoost	83.7	LD	76.4	50.3	60.7
			SU	85.1	94.8	89.7
Foraging behavior-by-posture	Random Forest	**58.87**	RE_LD	30.6	15.2	20.3
			RE_SU	46.2	34	39.1
			RU_LD	50.9	52.6	51.8
			RU_SU	52.2	39.9	45.2
	XGBoost	58.78	RE_LD	47.6	10.1	16.7
			RE_SU	43.1	13.8	21
			RU_LD	52.3	55.5	53.8
			RU_SU	64.4	25.7	36.7

## Data Availability

The data presented in this study are openly available in USDA at DOI: 10.15482/USDA.ADC/28507400.

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
