# Peer review of "Multi-Sensor Integration and Machine Learning for High-Resolution Classification of Herbivore Foraging Behavior"

_animals, 2025, doi:10.3390/ani15070913_

Round 1
Reviewer 1 Report
Comments and Suggestions for Authors
The article is aimed at increasing the efficiency of livestock farming by identifying animal behavior patterns and making decisions based on them. I believe that this article is written on a relevant topic and can increase not only the efficiency of economic activity, but also food security.
The authors have completed a fairly comprehensive review of existing solutions, their efficiency and application.
The research methodology, equipment used, dataset structure, evaluation criteria and machine learning models used are described in sufficient detail.
The efficiency of the models considered has been assessed.
There are a number of comments on the work:
1. No information is provided on the required accuracy (performance indicators) of the models.
2. No assessment or forecast of the increase in the efficiency of the technological processes for which the models will be applied has been made.
3. The conclusions do not contain numerical data confirming the achieved result.
4. The choice of machine learning models used is not justified.
Author Response
Response to Reviewer 1 Comments
We sincerely appreciate your time and effort in reviewing our manuscript. Your insightful feedback has been significant for improving the clarity and quality of our work. Below, we provide detailed responses to each of your comments and outline the corresponding revisions.
Point-by-Point Response to Comments and Suggestions for Authors
Comment 1:
Q1: No information is provided on the required accuracy (performance indicators) of the models.
Response:
Thank you for your observation. The accuracy and performance indicators of the models are presented in the results section, ranging from 83.9% to 58.8%. These values are first introduced in the abstract (line 29 – 37) and then detailed within the manuscript. Specifically:
- State classification: Reported in lines 374–388 (Random Test-Split) and lines 414–420 (Cross-Validation).
- Activity classification and activity-by-posture classification: Discussed in lines 394–404 (Random Test-Split) and lines 443–454 (Cross-Validation).
Comment 2:
Q2: No assessment or forecast of the increase in the efficiency of the technological processes for which the models will be applied has been made.
Response:
We appreciate your feedback. The manuscript discusses the relevance of the data partition strategy and the selected models for both general and complex foraging classification. This is highlighted in the abstract (lines 39–41):
"These results emphasize the reliability of the Cross-Validation (CV) data partition strategy in improving the Random Forest (RF) model’s ability to manage complex behavioral patterns. Additionally, they underscore the importance of continuous recording devices and movement metrics for accurately monitoring cattle behavior."
Furthermore, a detailed discussion on how classification accuracy influences the efficiency of the technological process is presented in the results section (lines 515–710).
Comment 3:
Q3: The conclusions do not contain numerical data confirming the achieved result.
Response:
Thank you for your suggestion. We have revised the conclusion to include numerical accuracy values, ensuring that the achieved results are clearly supported by data (lines 718–722).
Comment 4:
Q4: The choice of machine learning models used is not justified.
Response:
We appreciate this feedback and have revised the manuscript to clarify the selection of machine learning models. Specifically, we have updated Section 2.5, titled "Data Partition Strategy and Machine Learning Models" (starting at line 303), to ensure smooth readability and clear understanding of the subsection concerning the choice of models. This subsection first defines all the models and then provides a detailed explanation of the rationale behind their selection in this study (lines 312–322).
Thank you again for your thoughtful comments. We believe these revisions enhance the clarity and rigor of our manuscript, and we appreciate your constructive input.
Reviewer 2 Report
Comments and Suggestions for Authors
Dear Authors,
The manuscript discusses the application of various methods and techniques for classifying the behavior of herbivores when searching for food for herbivorous mammals. This is of practical interest. However, the manuscript cannot be published in this form. The text of the manuscript is mixed up in different chapters and should be moved to the appropriate chapters. In the Discussion, the authors often repeat that their results are consistent with the results of other researchers, but they forget to show significant differences. In many methodological aspects, the authors omit important information. It should be added. There are questions about separating grazing and walking. Cows often alternate between these states, so it is difficult to determine when this state actually occurs. In addition, animals often chew during rest, so it is also difficult to separate here. The authors divided the time interval of the day by hours, but a more accurate division is given by intervals from sunrise to sunset. This varies by season. The focus should be on writing the conclusion of the manuscript based on the implementation of practical recommendations of the results. The authors have obtained interesting results, presented them and based on the proposed hypothesis they should be disclosed in the conclusions of the manuscript. After all comments have been eliminated, the manuscript can be reviewed again.

Dear Editor,
The manuscript discusses the application of various methods and techniques for classifying the behavior of herbivores when searching for food for herbivorous mammals. This is of practical interest. However, the manuscript cannot be published in this form. The text of the manuscript is mixed up in different chapters and should be moved to the appropriate chapters. In the Discussion, the authors often repeat that their results are consistent with the results of other researchers, but they forget to show significant differences. In many methodological aspects, the authors omit important information. It should be added. There are questions about separating grazing and walking. Cows often alternate between these states, so it is difficult to determine when this state actually occurs. In addition, animals often chew during rest, so it is also difficult to separate here. The authors divided the time interval of the day by hours, but a more accurate division is given by intervals from sunrise to sunset. This varies by season. The focus should be on writing the conclusion of the manuscript based on the implementation of practical recommendations of the results. The authors have obtained interesting results, presented them and based on the proposed hypothesis they should be disclosed in the conclusions of the manuscript. After all comments have been eliminated, the manuscript can be reviewed again.
Author Response
Response to Reviewer 2 Comments
- Summary
Thank you for taking the time to review this manuscript. Please find our detailed responses below, along with the corresponding revisions. We greatly appreciate your feedback and have made revisions based on your suggestions. - Point-by-Point Response to Comments and Suggestions for Authors
Summary Comments (Lines 15–25)
Comment 1 (Lines 16–17): "Its interconnected"—be more detailed.
Response to Comment 1:
- Thank you for your observation. The sentence distinguishes between posture (position), activity (active vs. static), and foraging behaviors (i.e., rumination, grazing, resting, walking). We modified to: “The study used Random Test-Split (RTS) and Cross-Validation (CV) machine learning data partition methods to test different models to classify cattle behavior, including activity and posture states, and foraging behaviors, using GPS coupled accelerometer data.”
- Additionally, the term "specific sensor" refers to key predictors derived from GPS and accelerometer data, which are combined. This flexibility allows ranchers to use either or both sensors to monitor specific behaviors based on their management needs. Thus, we opted to retain the word (line 22).
- As suggested, we have removed the phrase 'improve animal welfare' (line 25).
Abstract Comments
Comment 2 (Line 41): Replace "movement metrics"
Response to Comment 2 (line 41).
Thank you for your suggestion. We have made the following revisions:
• "Movement metrics" has been revised to "movement data" (line 41).
Comment 3 (Line 43): Replace "foraging behaviors"
Response to Comment 3 (line 44):
• "Foraging behaviors" has been changed to "behavior classification" (line 44).
Introduction Comments
Comment 4 (Line 46): Specify which ones (indicate in brackets).
Response to Comment 4 (line 46).
The word "(cattle)" has been added in brackets (line 46).
Comment 5 (Line 59): Add: "It is known that this subsequently changes the living conditions of wild animals [Andreychev et al. 2015, Afonso et al. 2021]."
Response to Comment 5 (lines 59–60) :
The suggested sentence has been included (lines 59–60).
Comment 6 (Line 65): Move "labor-intensive" to the discussion.
Response to Comment 6:
The term "labor-intensive" has been moved from line (66) to the discussion (line 644).
Comment 7 (Lines 67–85): Move to the discussion.
Response to Comment 7 (lines 631–643):
The paragraph in lines 67–85 has been retained for context and clarity. However, key points are further elaborated in the discussion section (lines 631–643).
Comment 8 (Lines 123–126): Move the hypothesis above the research aim.
Response to Comment 8(lines 123–126) :
The hypothesis (lines 123–126) has been moved above the research aim (now lines 115–118).
Materials and Methods
Comment 9 (Line 131): Describe the soils.
Response to Comment 9 (line 133).
Clay loam soil type was added (line 133).
Comment 10 (Lines 133–134): Write the full names of species (author, year) at the first mention in the text.
Response to Comment 10 (line 135-136). :
These are grass species in full name; we provided common names and scientific names within brackets. Thus, we opted to retain this text (135-136).
Comment 11 (Line 142): Explain why the experiment was conducted from July to September 2024.
Response to Comment 11 (lines 144–148).
An explanation was added (lines 144–148).
Comment 12 (Line 145): Remove "randomly fitted" GPS.
Response to Comment 12(Line 151).:
The term "randomly fitted" was retained to clarify the non-biased fitting of GPS collars during assignment to different animals. (Line 151).
Comment 13 (Line 182): The distribution of the data by sunrise and sunset times is important. Why didn't you use this additionally in the methods?
Response to Comment 13:
Thank you for your valuable suggestion. We appreciate the reviewer highlighting the importance of incorporating sunrise and sunset times in the analysis. However, these times were well captured, and the rationale behind our choice of observation intervals is explained below (lines 188-189). In this study, we chose to focus on fixed observation intervals: morning (7:00 am–11:00 am), afternoon (12:00 pm–4:00 pm), and evening (5:00 pm–8:00 pm), based on the assumption that they would capture a broad range of activities across daylight hours. These intervals were intentionally chosen to align closely with the natural light conditions during the summer months in Logan, UT, where sunrise occurs around 6:00 AM and sunset around 9:00 PM. The term "daylight hours" in the justification statement for the selected times means we are covering less than a 1-hour difference before and after the actual sunrise and sunset times, during which we were able to effectively observe and record the animals during observations. Moreover, our study focused on machine learning, and the observation hours were used as a training dataset designed to predict the overall activities of the animals over 24 hours. This comprehensive approach allows for the development of models that can recognize and predict animal behaviors throughout the entire day. Several studies, including those by Augustine et al. (reference in text), have typically used random observations of 5 to 6 hours as ground truth data for behavior classification, which is considerably shorter than the continuous 12-hour approach we employed.
Comment 14 (Line 185): Grazing and walking have a lot in common to some extent.
Response to Comment 14 line 227- 238:
Thank you for your valuable comment. Our study used video cameras that allowed to clearly differentiate between these two activities and then calibrate accelerometer data accordingly. We also removed line 185–186, which mentioned these activities, to avoid repetition as suggested in line 219 and opted to leave these activities in sub section 2.3 line 227- 238 where they are clearly defined.
Comment 15 (Line 218): Why exactly three specialists?
Response to Comment 15 line 223:
Three specialists on animal behavior activities were involved to train technicians on how to accurately label activities in the datasheet while reviewing the video footage. This ensured consistency and reliability in the labeling process. Hence, we opted to retain the word at line 223.
Results
Comment 16 (Lines 334–340) : I recommend presenting this in the form of a diagram for clarity.
Response to Comment 16 Line 343 - 361:
Thank you for the recommendation. We have included a diagram for clarity. Line 343 - 361
Comment 17 (Lines 340–342): It should be added whether there are differences in the sample depending on the age of the animals.
Response to Comment 17 (lines 148–150).:
We have added a clarification (line 336) to indicate that the observed cows were mother cows of similar age and weight (lines 148–150).
Comment 18 (Line 367): There is no need to highlight this separately. It is enough to indicate the table number in brackets at the end of the text.
Response to Comment 18 (lines 395 and 402).:
We have revised the text to indicate the table number in brackets (lines 395 and 402).
Comment 19 (Line 385): It is better to present it in a graphical form.
Response to Comment 19 (line 410 -411).:
Thank you for the suggestion. We believe it is important to keep this information in table form, as presenting it graphically may create busy figures. Thus, we opted to retain the table. (line 410 -411).
Comment 20 (Line 414): These are methods.
Response to Comment 20 line 439:
We have omitted the sentence (line 439) to avoid redundancy.
Discussion
Comments 21 line 489-491: What is the difference between these studies and yours?
Response to Comment 21 (Line 524 – 533)
We appreciate the reviewer’s suggestion to clarify the differences between our study and the findings of Chakraborty et al. and Wyner et al. In our revised manuscript, we have explicitly distinguished our work from these studies (Line 524 - 533).
Comments 22 line 533-535: What is the difference between these studies and yours?
Response to Comment 22 (Line 582 - 590):
We appreciate the reviewer’s suggestion to clarify the differences between our study and the findings of Wang’s et al In our revised manuscript, we have explicitly distinguished our work from these studies ( Line 582 - 590).
Comments 23 line 544-547: What is the difference between these studies and yours?
Response to Comment 23 (Line 602-613):
We appreciate the reviewer’s suggestion to clarify the differences between our study and the findings of Pütün and Yilmaz [44].In our revised manuscript, we have explicitly distinguished our work from these studies ( Line 602-613).
Conclusion;
Comments 24 line 664-669: Where are your results already applicable? What recommendations have you made and to whom? This will improve the practical significance of the study.
Response to Reviewer Comment 24 (Lines 732-746):
Thank you for your comment. We have clarified the practical applications and recommendations (Lines 732-746). The findings are applicable to livestock management, especially in pasture-based systems, where accurately classifying behaviors like grazing, ruminating, and resting is essential for optimizing resource allocation and animal welfare. Machine learning methods like Random Forest and XGBoost can provide insights into animal health and productivity, improving decision-making processes during grazing. We recommend ranchers and land managers adopt these methods, particularly in systems with GPS and accelerometer data, to improve behavioral monitoring, grazing management, and resource distribution. Additionally, applying data-driven strategies to monitor foraging behaviors can help prevent overgrazing and optimize pasture use. Further research should address challenges like data imbalances and sensor fix intervals to improve accuracy. By following these recommendations, ranchers and land managers can enhance operational efficiency and animal welfare.
Thank you again for your thoughtful comments. We believe these revisions enhance the clarity and rigor of our manuscript, and we appreciate your constructive input.
Round 2
Reviewer 2 Report
Comments and Suggestions for Authors
Dear Authors,
I am satisfied with the revision of the manuscript. The manuscript has been corrected and additional data have been added. The emphasis is on the practical application of the research results. This is important. Monitoring cattle foraging behavior using data-driven strategies can optimize pasture use and prevent overgrazing. The review of the research results was chosen appropriately, as were the statistical methods used for its analysis. The article takes into account the comments on the methodology. The analysis and conclusion for each chapter are sufficient and do not raise objections. References to sources of literature have been adjusted. The results of previous studies by other authors have been taken into account. I recommend it for the journal Animals.
Author Response
Response to Reviewer 2 Comments
We sincerely appreciate your time and effort in reviewing our manuscript. Your insightful feedback has been significant for improving the clarity and quality of our work. Below, we provide detailed responses to each of your comments and outline the corresponding revisions.
Point-by-Point Response to Comments and Suggestions for Authors
Comments 1 : I am satisfied with the revision of the manuscript. The manuscript has been corrected and additional data have been added. The emphasis is on the practical application of the research results. This is important. Monitoring cattle foraging behavior using data-driven strategies can optimize pasture use and prevent overgrazing. The review of the research results was chosen appropriately, as were the statistical methods used for its analysis. The article considers the comments on the methodology. The analysis and conclusion for each chapter are sufficient and do not raise objections. References to sources of literature have been adjusted. The results of previous studies by other authors have been considered. I recommend it for the journal Animal
Response to Comment 1:
We appreciate the reviewer’s positive feedback and recommendation for publication in Animal. We are pleased that the revisions, including additional data and refined methodology, have strengthened the manuscript. The emphasis on the practical applications of data-driven strategies for monitoring cattle foraging behavior and optimizing pasture use aligns with the study’s objectives. We also acknowledge the reviewer’s recognition of the statistical methods, analysis and conclusions, and adjustments to references, ensuring alignment with previous research. Thank you for your thorough review and valuable input.